# Rapid antigen testing as a reactive response to surges in nosocomial SARS-CoV-2 outbreak risk

David R. M. Smith 1,2,3,8✉, Audrey Duval1,2,4,8, Jean Ralph Zahar4,5, the EMAE-MESuRS Working Group on Nosocomial SARS-CoV-2 Modelling*, Lulla Opatowski1,2 & Laura Temime3,6

Healthcare facilities are vulnerable to SARS-CoV-2 introductions and subsequent nosocomial outbreaks. Antigen rapid diagnostic testing (Ag-RDT) is widely used for population screening, but its health and economic benefits as a reactive response to local surges in outbreak risk are unclear. We simulate SARS-CoV-2 transmission in a long-term care hospital with varying COVID-19 containment measures in place (social distancing, face masks, vaccination). Across scenarios, nosocomial incidence is reduced by up to 40-47% (range of means) with routine symptomatic RT-PCR testing, 59-63% with the addition of a timely round of Ag-RDT screening, and 69-75% with well-timed two-round screening. For the latter, a delay of 4-5 days between the two screening rounds is optimal for transmission prevention. Screening efficacy varies depending on test sensitivity, test type, subpopulations targeted, and community incidence. Efficiency, however, varies primarily depending on underlying outbreak risk, with health-economic benefits scaling by orders of magnitude depending on the COVID-19 containment measures in place.

[1] Institut Pasteur, Epidemiology and Modelling of Antibiotic Evasion (EMAE), Paris, France. [2] Université Paris-Saclay, UVSQ, Inserm, CESP, Anti-infective evasion and pharmacoepidemiology team, Montigny-Le-Bretonneux, France. [3] Modélisation, épidémiologie et surveillance des risques sanitaires (MESuRS), Conservatoire national des arts et métiers, Paris, France. [4] IAME, UMR 1137, Université Paris 13, Sorbonne Paris Cité, Paris, France. [5] Service de Microbiologie Clinique et Unité de Contrôle et de Prévention du Risque Infectieux, Groupe Hospitalier Paris Seine Saint-Denis, AP-HP, Bobigny, France. [6] PACRI unit, Institut Pasteur, Conservatoire national des arts et métiers, Paris, France. [8] These authors contributed equally: David R. M. Smith, Audrey Duval. *A list of authors and their affiliations appears at the end of the paper. ✉email: david.smith@pasteur.fr

A range of vaccines have proven safe and effective for the prevention of SARS-CoV-2 infection, offering hope towards an end to the COVID-19 pandemic[1–3]. Unfortunately, despite high vaccination coverage, hospitals and long-term care facilities (LTCFs) remain vulnerable to nosocomial outbreaks[4]. LTCFs globally report instances of breakthrough infection and ensuing transmission among immunized staff and residents. This is notably due to variants of concern like B.1.1.7 (Alpha), B.1.351 (Beta) and B.1.617.2 (Delta), which may partly escape vaccine-induced immunity relative to wild type[5–8]. This suggests that testing and screening interventions will remain important tools for detecting and isolating SARS-CoV-2 infections in healthcare facilities, even in settings with high vaccine uptake.

However, while repeated screening may be an effective tool for nosocomial transmission prevention[9,10], it also imposes substantial economic cost and occupational burden on healthcare staff[11,12]. For potentially vulnerable, resource-limited facilities, a key challenge is knowing if, when and how to implement SARS-CoV-2 surveillance interventions[13]. When outbreak risk is low – perhaps in a highly immunized LTCF around low community incidence and few variants of concern – screening at frequent intervals is probably an inefficient use of limited health-economic resources.

Yet outbreak risk is in constant flux, and is sometimes predictable. Festive holidays, for instance, draw individuals from distant places into close contact for prolonged periods, and have been associated with surges in SARS-CoV-2 epidemic risk in China, Israel, and elsewhere[14,15]. Into 2022, widespread post-holiday, inter-generational population movement in the context of variants like Delta and B.1.1.529 (Omicron) may pose similar concerns[16]. In such a context where local knowledge or epidemiological data indicate a suspected spike in epidemic risk, or where identification of a new case or exposed contact within a healthcare facility indicates potential for a nosocomial outbreak, reactive use of antigen rapid diagnostic testing (Ag-RDT) may be an efficient public health response.

Here, we aim to help determine the best surveillance strategies for control of SARS-CoV-2 transmission in healthcare facilities in the context of a surge in nosocomial outbreak risk. To this end, we adapt a simulation model and assess the epidemiological efficacy and health-economic efficiency of single or repeated Ag-RDT screening coupled with routine symptomatic reverse transcriptase polymerase chain reaction (RT-PCR) testing. Simulated Ag-RDT screening interventions are conceptualized as reactive public health responses, conducted in a long-term care hospital with varying COVID-19 containment measures in place.

## Results

**SARS-CoV-2 outbreak risk depends on the COVID-19 prevention measures in place.** Following a simulated surge in SARS-CoV-2 outbreak risk, nosocomial incidence varied across LTCFs depending on the COVID-19 containment measures in place (Fig. 1). Low-control LTCF 1 experienced exponential epidemic growth driven by patient-dominated clusters, by two weeks reaching a mean cumulative number of incident nosocomial SARS-CoV-2 infections I= 28.9 (range 0–82). With patient social distancing in the moderate-control LTCF 2, epidemic growth was linear, and nosocomial incidence was reduced by a mean 62.2% relative to LTCF 1, with a similar share of infections among patients and staff (Supplementary Fig. S2). Finally, with vaccination, mandatory face masks and social distancing combined in the high-control LTCF 3, outbreaks tended towards extinction, with a mean 96.2% reduction in incidence relative to LTCF 1. In this last LTCF, staff members infected in the

community represented the majority of cases, and rarely infected others in the hospital.

Super-spreaders drove high incidence in LTCF 1 (representing a mean 5.5% of infected individuals, but responsible for 47.3% of nosocomial infections) but less so in LTCFs with more robust COVID-19 containment measures (3.1% and 23.4% in LTCF 2; 0.2% and 1.1% in LTCF 3; Supplementary Fig. S3). In a sensitivity analysis evaluating outbreak risk across asymmetric patient-staff vaccine coverage, patient more than staff immunization was impactful for preventing patient transmission to other individuals (see Supplementary Fig. S4). Conversely, patient and staff immunization were similarly impactful against staff transmission. Given low rates of patient and/or staff immunization, this setting was nonetheless resilient to outbreaks when alternative containment measures were also in place (i.e. with social distancing and mandatory face masks).

**Reactive Ag-RDT screening complements, but does not replace routine RT-PCR testing.** Surveillance interventions were evaluated in each LTCF for their ability to prevent SARS-CoV-2 transmission, and surveillance efficacy ($E$) is reported as the mean (95% CI) relative reduction in I due to surveillance. Routine RT-PCR testing significantly reduced incidence of hospital-acquired SARS-CoV-2 infection, by $E = 39.8\%$ (39.1–40.3%) in LTCF 1, $E = 41.2\%$ (40.5–41.9%) in LTCF 2, and $E = 46.6\%$ (45.4–47.5%) in LTCF 3 (Fig. 2, Supplementary Fig. S7). This corresponded to a mean 11.9 infections averted in LTCF 1, 4.8 in LTCF 2, and 0.51 in LTCF 3 (Supplementary Fig. S8). Greater relative efficacy in higher-control LTCFs was consistent with a higher average probability of positive test results, a consequence of fewer new, as-yet undetectable infections (Supplementary Fig. S6). On its own, 1-round Ag-RDT screening was less effective than routine testing, reducing the incidence of hospital-acquired SARS-CoV-2 infection by up to $E = 31.2–37.5\%$ (range of means across LTCFs when conducted on day 1). For 1-round Ag-RDT screening in combination with routine testing, nosocomial incidence was reduced by up to $E = 58.4–63.5\%$. Among infections not prevented by routine testing, this represents a marginal $E_m = 30.5–32.4\%$ reduction in remaining incidence due to screening. Whether paired with routine testing or conducted independently, more immediate 1-round Ag-RDT screening was generally more effective (Fig. 2).

**Two-round Ag-RDT screening improves screening efficacy, but is time-sensitive.** Two-round screening—conducting a first round of screening immediately upon outbreak detection, and an additional second round over the following days—increased surveillance efficacy. Nosocomial incidence was reduced by up to $E = 69.4–75.0\%$ across LTCFs with well-timed 2-round screening (Fig. 2). This represents a marginal reduction of $E_m = 48.1–52.8\%$ among remaining infections not averted by routine testing alone. Optimal timing for the second round of screening was on days 5–6 (4–5 days after the first round). In an alternative scenario of higher community incidence and more frequent introductions of SARS-CoV-2 into the LTCF, screening was overall less effective for transmission prevention than in the baseline scenario, and optimal timing for the second screening round was delayed further in LTCFs 2 and 3 (Supplementary Fig. S9c).

**Screening efficacy depends on screening targets and test type.** Targeting both patients and staff for screening was always more effective than only targeting one or the other (Supplementary Fig. S9a). Targeting only patients was substantially more effective than staff for LTCF 1, consistent with its large patient-led outbreaks. This difference was less pronounced in LTCF 2, while in LTCF 3 screening efficacy was nearly identical whether targeting

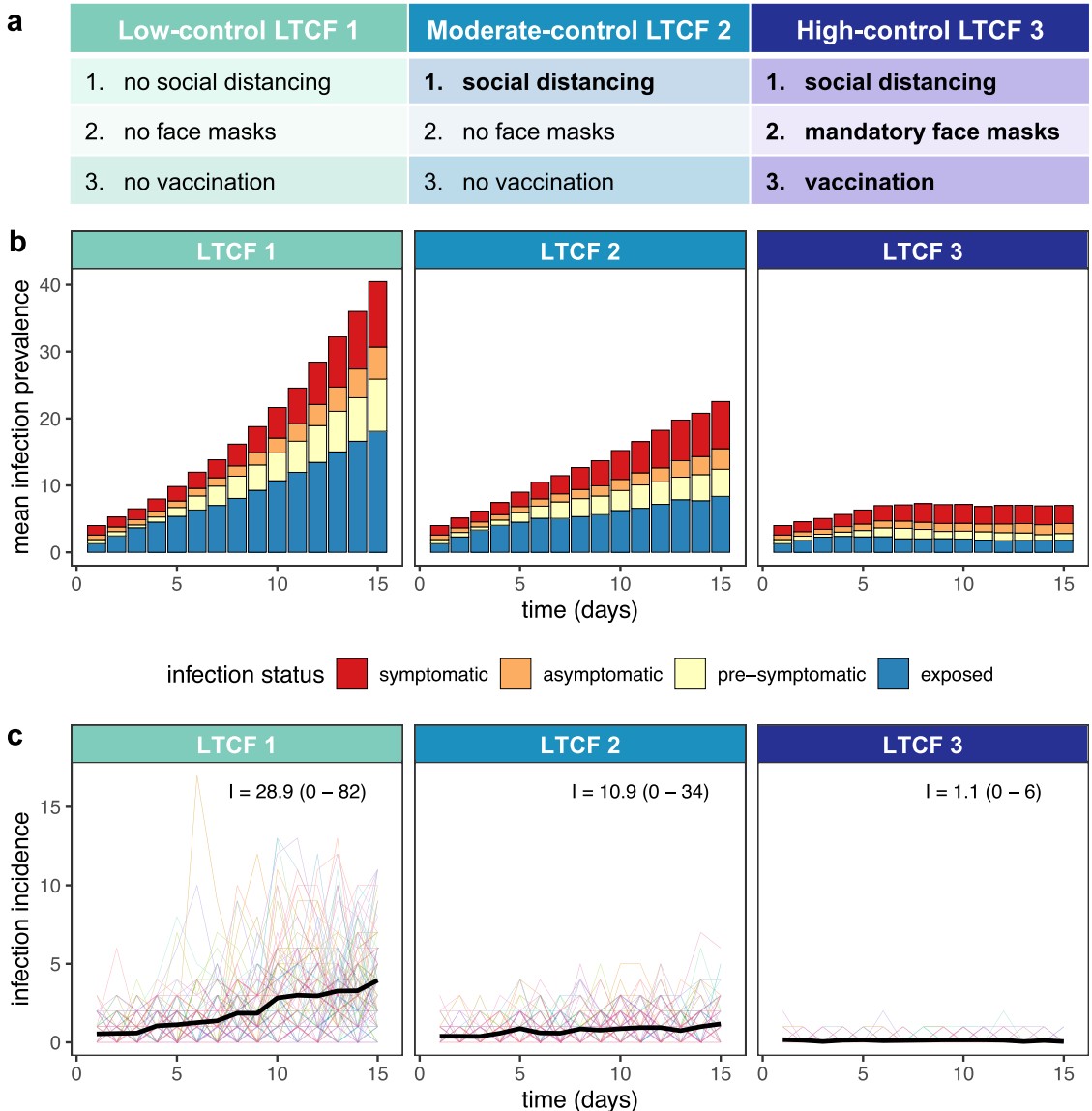

**Fig. 1 Modelling context: simulating SARS-CoV-2 outbreaks in a long-term care facility (LTCF) with three different levels of COVID-19 control. a** A list of the COVID-19 containment measures in place across low-control LTCF 1, moderate-control LTCF 2, and high-control LTCF 3 (see Supplementary information section I for details). **b** Daily infection prevalence, the mean number of individuals in each infection stage (colours) over time. Pre-symptomatic infection combines pre-symptomatic and pre-asymptomatic infection, and symptomatic infection combines mild symptomatic and severe symptomatic infection. **c** Daily nosocomial infection incidence, the number of new SARS-CoV-2 infections acquired within the LTCF each day. Thin coloured lines are individual simulations; the thick black line is the mean across 100 simulations. In text, the mean (range) cumulative nosocomial incidence, I, over two weeks. The proportions of simulations with ≥1 cumulative nosocomial cases were 98%, 98% and 59% in LTCFs 1, 2 and 3, respectively; the proportions with ≥5 cumulative nosocomial cases were 91%, 73% and 2%; the proportions with ≥10 cumulative nosocomial cases were 82%, 55% and 0%; and the proportions with ≥25 cumulative nosocomial cases were 51%, 6% and 0%.

patients or staff. We also evaluated the use of RT-PCR instead of Ag-RDT for screening, maintaining its higher diagnostic sensitivity and longer turnaround time (24 h). For all types of screening considered (1-round, 1-round with routine testing, 2-round with routine testing), Ag-RDT screening was more effective for transmission prevention than RT-PCR screening, suggesting that faster turnaround time for Ag-RDT outweighs its reduced sensitivity. This finding was robust to a sensitivity analysis considering an alternative curve for diagnostic sensitivity of Ag-RDT relative to RT-PCR (Supplementary Fig. S9b).

**Screening efficiency and cost-effectiveness scale with underlying outbreak risk.** The efficiency of simulated surveillance

interventions was evaluated using three distinct outcomes: *apparent efficiency* (A, the average expected number of infections detected per test used), *real efficiency* (R, the average number of infections prevented per test used), and the *cost-effectiveness ratio* (CER, testing unit costs per infection prevented). The efficiency of routine RT-PCR testing varied substantially across LTCFs: mean apparent efficiency ranged from $A = 28–65$ cases detected/1000 RT-PCR tests, while mean real efficiency ranged from $R = 5–105$ cases averted/1000 RT-PCR tests (Supplementary Fig. S10). Relative to RT-PCR, the apparent efficiency of Ag-RDT screening interventions was similar across LTCFs. For example, for the most effective surveillance intervention (routine RT-PCR testing + 2-round Ag-RDT screening on days 1 and 5; intervention #23 in Supplementary table S2), apparent efficiency of screening

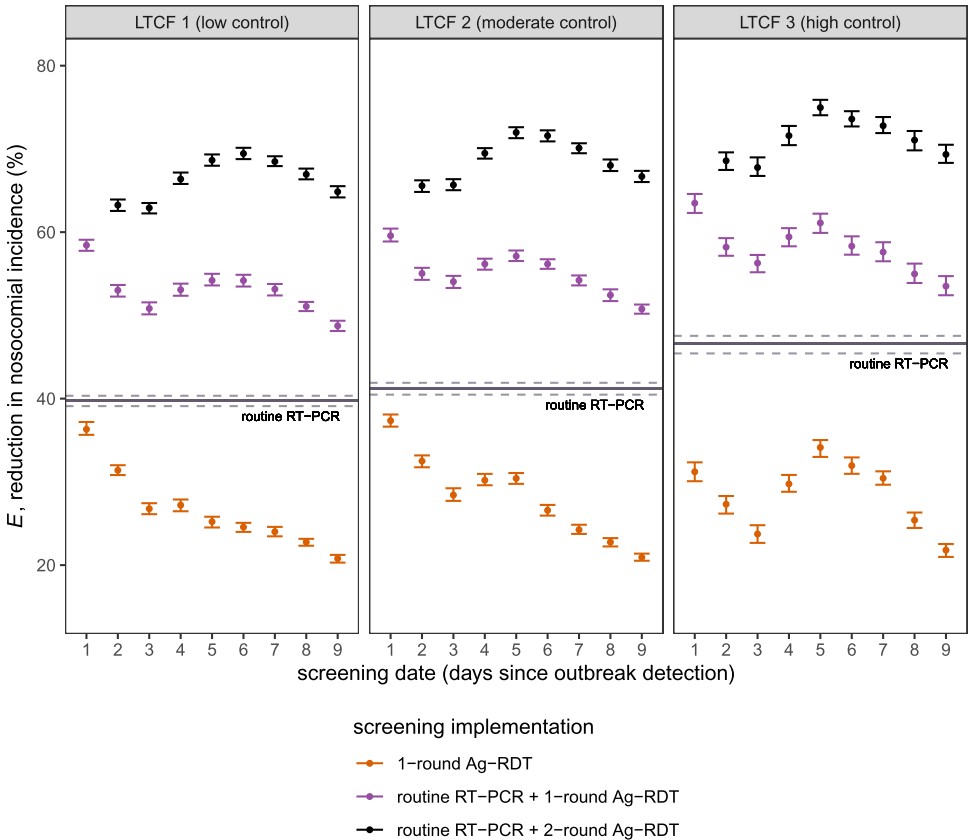

**Fig. 2 Efficacy of Ag-RDT screening interventions for reducing nosocomial SARS-CoV-2 incidence.** Points represent mean efficacy (across $n =$ 10,000 simulations) for each of 26 screening interventions, arranged by timing of the screening intervention (days since initial outbreak detection, *x*-axis) and coloured by screening implementation (either as 1-round screening with no other testing, orange; as 1-round screening in combination with routine RT-PCR testing, purple; or as 2-round screening with routine RT-PCR testing, black). For 2-round screening, the first round was conducted on day 1, with points arranged according to the date of the second round (days 2–9). The solid horizontal line corresponds to mean efficacy of routine RT-PCR testing in absence of screening, which is conducted continuously over time and does not correspond to a specific date. Relative reductions in incidence were similar across LTCFs, but there was significant variation in the number of infections averted (Supplementary Fig. S8). Error bars (dashed lines for routine testing) correspond to 95% confidence intervals estimated by bootstrap resampling ($n = 10,000$). Baseline assumptions underlying simulations include: "low" community SARS-CoV-2 incidence; time-varying Ag-RDT sensitivity relative to RT-PCR (Ag-RDT A); and screening interventions that target all patients and staff in the LTCF. RT-PCR = reverse transcriptase polymerase chain reaction; Ag-RDT = antigen rapid diagnostic testing; LTCF = long-term care facility.

ranged from $A = 3.3$–$3.6$ infections detected/1000 Ag-RDT tests if targeting patients, $A = 6.2$–$6.3$ infections detected/1000 Ag-RDT tests if targeting staff, and $A = 5.1$–$5.2$ infections detected/1000 Ag-RDT tests if targeting both patients and staff. This reflects that screening interventions detected similar numbers of infections in each LTCF relative to the large number of tests used. However, LTCFs varied greatly in terms of real health-economic benefits of screening. For example, for this same intervention (#23) targeting patients, marginal real efficiency of screening was $R_m = 19.6$ cases averted/1000 Ag-RDT tests in LTCF 1, $R_m = 5.3$ cases averted/1000 Ag-RDT tests in LTCF 2, and $R_m = 0.5$ cases averted/1000 Ag-RDT tests in LTCF 3 (Fig. 3). Efficiency and other measures of screening performance (TPV, NPV, PPV, NPV) varied substantially over time, depending on which populations were targeted by screening (Supplementary Fig. S11).

Cost-effectiveness of surveillance interventions varied by orders of magnitude across LTCFs (Fig. 4). In LTCF 1, assuming baseline per-test unit costs (€50/RT-PCR test, €5/Ag-RDT test), the cost-effectiveness ratio of the most effective surveillance intervention (#23) was CER = €469 (95% CI: €462–€478)/case averted (Fig. 4). In LTCF 2, the same intervention cost CER = €1180 (€1166–€1200)/case averted, and in LTCF 3 CER= €11,112

(€10,825–€11,419)/case averted. Cost-effectiveness ratios were similar whether conducting one or two rounds of Ag-RDT screening (Supplementary Fig. S12). Overall, for interventions combining routine testing and reactive screening, cost-effectiveness ratios were more sensitive to costs of Ag-RDT screening tests than routine RT-PCR tests (Fig. 4). At a fixed unit cost of €50/RT-PCR test in the high-control LTCF 3, cost-effectiveness ratios for intervention #23 were approximately: CER = €16,000/case averted at €10/Ag-RDT test, CER = €30,000/case averted at €25/Ag-RDT test, and CER = €54,000/case averted at €50/Ag-RDT test. Conversely, in the low-control LTCF, cost-effectiveness ratios remained below CER= €5000/case averted up to €100/Ag-RDT test. When reactive Ag-RDT screening and routine RT-PCR testing were considered separately as independent surveillance strategies, routine testing was always more cost-effective than reactive screening per € spent on surveillance costs (Supplementary Fig. S13).

## Discussion

Surges in nosocomial SARS-CoV-2 outbreak risk are often predictable, resulting from phenomena like local emergence of a

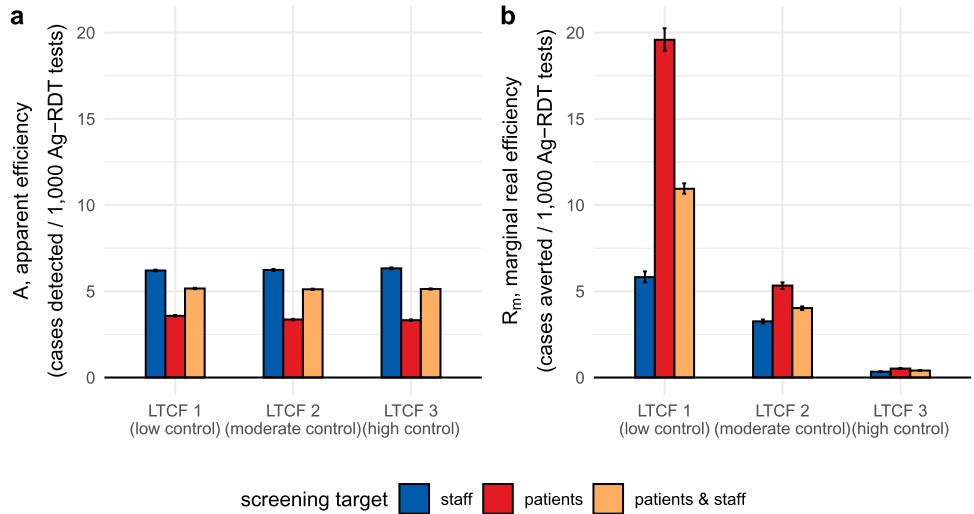

**Fig. 3 Efficiency of Ag-RDT screening: infection detection versus infection prevention.** Efficiency of Ag-RDT screening in the context of a highly effective surveillance strategy (intervention #23, routine RT-PCR testing + 2-round Ag-RDT screening on days 1 and 5), comparing (**a**) apparent screening efficiency with (**b**) marginal real screening efficiency. Marginal real screening efficiency describes efficiency of Ag-RDT screening for prevention of remaining nosocomial SARS-CoV-2 infections not already averted by routine RT-PCR testing. Screening interventions targeted either all members of staff (blue), all patients (red), or all individuals in the LTCF (orange). Baseline assumptions underlying simulations include: "low" community SARS-CoV-2 incidence and time-varying Ag-RDT sensitivity relative to RT-PCR (Ag-RDT A). Bar heights and error bars correspond to means and 95% confidence intervals estimated by bootstrap resampling (n = 10,000). RT-PCR = reverse transcriptase polymerase chain reaction; Ag-RDT = antigen rapid diagnostic testing; LTCF = long-term care facility.

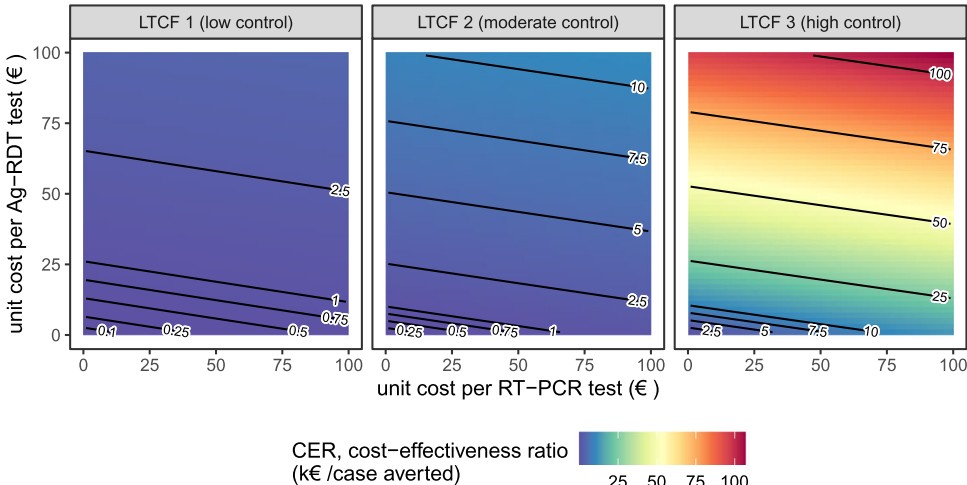

**Fig. 4 Surveillance cost-effectiveness: underlying outbreak risk outweighs testing unit costs.** Mean cost-effectiveness ratios for a highly epidemiologically effective surveillance strategy (intervention #23, routine RT-PCR testing + 2-round Ag-RDT screening on days 1 and 5), estimated as testing unit costs per infection averted while varying unit costs for RT-PCR tests (x-axis) and Ag-RDT tests (y-axis). Baseline assumptions underlying simulations include: "low" community SARS-CoV-2 incidence; time-varying Ag-RDT sensitivity relative to RT-PCR (Ag-RDT A); and screening interventions that target all patients and staff in the LTCF. RT-PCR = reverse transcriptase polymerase chain reaction; Ag-RDT = antigen rapid diagnostic testing; LTCF = long-term care facility.

highly transmissible variant, seasonal or festive gatherings that increase population mixing, and the identification of index cases or exposed contacts in a healthcare facility. When such risks are known, implementing reactive surveillance may help to identify and isolate asymptomatic and pre-symptomatic infections, limiting onward nosocomial transmission. Using simulation modelling, we demonstrate how reactive Ag-RDT screening complements routine RT-PCR testing in reducing nosocomial SARS-CoV-2 incidence following a known surge in outbreak risk. With two rounds of well-timed Ag-RDT screening, up to 75% of infections were prevented, compared to 47% with routine RT-PCR testing alone. Underlying outbreak risk was the greatest driver of screening efficiency, more important than screening

timing (immediate vs. delayed), test type (Ag-RDT vs. RT-PCR) or target (patients vs. staff). We estimated that a vulnerable LTCF gains between one and two orders of magnitude more health-economic benefit (>10 infections averted/1000 Ag-RDT tests used) than a resilient LTCF with alternative COVID-19 control measures already in place (<1 infection averted/1000 Ag-RDT tests).

Ag-RDT screening is widely used in healthcare settings, but there is limited empirical evidence demonstrating efficacy for SARS-CoV-2 transmission prevention[17]. Despite a range of studies reporting efficacy for case identification[18–20], interventional trials are needed to understand impacts on nosocomial spread. Our comparison of apparent and real screening efficiency

demonstrates why case identification may be a poor proxy measure for actual health and economic benefit. In the absence of empirical data, mathematical models have been useful tools to evaluate the performance of SARS-CoV-2 screening interventions in healthcare settings. Most studies have simulated use of routine screening at regular intervals (e.g. weekly, biweekly), finding that more frequent screening reduces outbreak probability, that targeting patients versus staff can significantly impact effectiveness, and that faster diagnostic turnaround time of Ag-RDT tends to outweigh reduced sensitivity relative to RT-PCR[9,10,21–28]. These conclusions were recapitulated in our findings.

Despite potential to reduce transmission, routine screening is an economic and occupational burden with uncertain suitability for low-risk healthcare settings[11,12]. These considerations have generally been neglected in previous work. A few modelling studies have estimated the cost-effectiveness of nosocomial screening interventions in specific use cases, including for hospital patients admitted with respiratory symptoms[29], patients admitted to German emergency rooms[30], and routine staff and resident testing in English nursing homes[31]. However, key impacts of stochastic transmission dynamics, screening heterogeneity, and other concomitant COVID-19 containment measures have rarely been accounted for, nor the wide range of unit costs for different testing technologies considered here. Further, to the best of our knowledge no studies have evaluated efficacy and efficiency of reactive, as opposed to routine screening, although findings from See et al. suggest greater efficiency of testing in outbreak versus non-outbreak settings[32]. Our work was further strengthened through the use of time-varying, test-specific diagnostic sensitivity (as opposed to time-invariant estimates often assumed in other work), facilitating assessment of optimal timing for multi-round screening. Overall, our use of high-resolution, stochastic, individual-based modelling complements previous studies in demonstrating how epidemiological and health-economic benefits of reactive screening scale with test sensitivity, screening timing, test type, population targets, and—most critically—underlying nosocomial outbreak risk.

This work focused on the detailed evaluation of surveillance interventions, explored in the context of three focal scenarios corresponding to varied adherence to standard COVID-19 containment measures (social distancing, face masks, vaccination). Each scenario resulted in fundamentally different epidemic dynamics (exponential growth, linear growth and extinction), allowing us to demonstrate how health-economic efficiency of surveillance varies with underlying nosocomial outbreak risk, while relative epidemiological efficacy is largely conserved. However, a detailed assessment of how different combinations of these other COVID-19 containment measures impact outbreak risk, transmission dynamics, surveillance efficacy and surveillance efficiency was beyond the scope of this work. To take the example of vaccination, outbreak risk (and hence surveillance efficiency) depends on, among other variables: the particular vaccine(s) used; their efficacy for prevention of transmission and disease in the context of locally circulating variants; the distribution of the number of doses/boosters received across patients and staff; and associated rates of immune waning and breakthrough infection. In a supplementary analysis we show how outbreak risk in our simulated LTCFs varies across asymmetric levels of immunizing seroprevalence (Supplementary Fig. S4), which may in turn impact optimal targets for screening (e.g. patient screening should likely be prioritized in a facility with disproportionately low patient or high staff vaccine coverage). For real-world facilities, local outbreak risk must be continuously assessed using local and up-to-date demographic, epidemiological and immunological data.

Our findings should be interpreted in the context of several methodological limitations. First, some results may reflect specificities of the rehabilitation hospital contact network and transmission process underlying our simulations. For instance, we estimated greater efficiency for screening patients relative to staff, but the opposite result may be expected in settings where staff have significantly higher rates of contact than patients. Further, the assumption that transmission risk saturates after 1 h of infectious contact may lack biological realism, but is unlikely to have substantially affected transmission dynamics (approximately 80% of contacts were <60 m in duration; median duration = 28.5 m). Second, our use of retrospective counterfactual analysis facilitated precise estimation of intervention efficacy, but precluded consideration of how surveillance interventions might impact human behaviour. For instance, healthcare workers that conduct screening inevitably come into contact with many individuals, potentially creating new opportunities for transmission. This risk may be mitigated through appropriate use of PPE during screening[33], and is not relevant if our results are interpreted in the context of self-administered auto-tests. Auto-testing may be a cost-effective intervention in the context of at-home testing in the community[34], but feasibility in healthcare settings is unclear, particularly for patients or residents among certain high-risk groups [35].

Third, counterfactual scenarios did not account for possible alternative infectors due to subsequent infectious exposures. For instance, an individual whose infection was averted due to isolation of their infector should nonetheless remain at risk of infection during subsequent contacts with other infectious individuals. This effect was likely negligible in higher-control LTCFs, where multiple acquisition routes are unlikely in the context of low nosocomial incidence, but may have resulted in overestimation of intervention efficacy in lower-control LTCFs. However, this should not have qualitatively changed our conclusions, as transmission chain pruning was conducted identically across interventions (e.g. testing vs. screening), and screening timing, test type (RT-PCR vs. Ag-RDT) and target (patients vs. staff). Fourth, our cost-effectiveness ratios only considered testing unit costs, but decision-makers must consider a range of other implementation costs, from human resources, to logistical coordination, to opportunity costs of false-positive isolation. Decision-makers may also have a wide variety of tests and manufacturers to choose from, including tests with heterogeneous sampling techniques (e.g. nasopharyngeal swabs vs. saliva or pharynx gargle samples), with potential consequences for surveillance costs, efficacy, compliance and occupational burden. (Note that the RT-PCR and Ag-RDT sensitivity curves used in the present work represent average results across a range of different tests used on upper respiratory specimens.) In particular, group testing (sample pooling) may be an efficient means of surveillance in low prevalence settings, reducing overall testing costs[36]. Finally, we limited our outbreak simulations to the two weeks following intervention implementation, implicitly assuming that LTCFs came to control nosocomial transmission at the same time. We thus do not capture potential downstream exponential benefits of preventing infections, including those that go on to seed transmission in the community.

Since its widespread uptake as a SARS-CoV-2 surveillance intervention, there has been substantial debate about whether the potential health-economic efficiency of Ag-RDT justifies an elevated risk of false-negative diagnosis[37,38]. Our findings are consistent with the view that Ag-RDT is on its own insufficient to eliminate nosocomial SARS-CoV-2 outbreak risk, but that it is nonetheless an effective component of multi-modal infection prevention strategies[39]. We demonstrate that reactive Ag-RDT screening is a potentially efficient public health response to surges in outbreak risk in the LTCF setting, but that its health and economic benefits scale by orders of magnitude depending on

other epidemiological risk factors, including the facility's inter-individual contact patterns, infection prevention measures, and vaccine coverage. This suggests that healthcare institutions should carefully evaluate their vulnerability to COVID-19—and hence potential returns on investment—before implementation of Ag-RDT screening interventions.

## Methods

**Ethical approval.** The inter-individual contact data used in this work was collected previously during the i-Bird study. The i-Bird study obtained all authorizations in accordance with French regulations regarding medical research and information processing. All French IRB-equivalent agencies accorded the i-Bird program official approval (CPP 08061; Afssaps 2008-A01284-51; CCTIRS 08.533; CNIL AT/YPA/SV/SN/GDP/AR091118 N°909036). Signed consent by patients and staff was not required according to the French Ethics Committee to which the project was submitted.

**Simulating SARS-CoV-2 outbreaks in the long-term care hospital setting.** We simulated SARS-CoV-2 outbreaks using CTCmodeler, a previously developed stochastic, individual-based transmission model in the LTCF setting[36,40]. Using high-resolution close-proximity interaction data from a 170-bed rehabilitation hospital in northern France, this model simulates (i) detailed inter-individual contacts among patients and staff, (ii) transmission of SARS-CoV-2 along simulated contact networks, and (iii) clinical progression of COVID-19 among infected individuals. More information about the model and underlying contact data are provided in Supplementary information section I.

A range of COVID-19 containment measures were built into the model. These include (i) a patient social distancing intervention (cancellation of social activities; see Supplementary Fig. S1), (ii) mandatory face masks among patients and staff (80% reduction in transmission rates), and (iii) imperfect vaccination of patients and staff (50% immunizing seroprevalence at simulation outset, compared to an assumed 20% baseline in scenarios without vaccination). This value is consistent with an estimated 53% efficacy of the mRNA BNT162b2 vaccine against infection with the Delta variant four months from second dose[41]. In a sensitivity analysis, we varied rates of immunizing seroprevalence from 0% to 100% across patients and staff to investigate potential epidemiological impacts of asymmetric immunization coverage. Three distinct combinations of containment measures were applied to the baseline LTCF to represent variable degrees of investment in COVID-19 prevention (Fig. 1a). These are presented as (i) *low-control LTCF 1*, with no explicit measures in place, (ii) *moderate-control LTCF 2*, with patient social distancing, and (iii) *high-control LTCF 3*, with patient social distancing, mandatory face masks and vaccination. Further modelling details are provided in Supplementary information section I.

**Simulation initialization.** Simulations were initialized to include a surge in SARS-CoV-2 outbreak risk, defined as a surge in SARS-CoV-2 introductions from the community. We assumed that 50% of patients and 100% of staff were exposed to contacts outside the LTCF in the week prior to simulation, conceptualized as representing family gatherings over a festive period. Calibrated to French epidemic data from January 2021, this translated to one patient and three staff infections, with a mean 1.4 symptomatic infections upon simulation initialization (Fig. 1b). Detection of symptomatic infection at simulation outset was interpreted as coinciding with initial SARS-CoV-2 outbreak detection within the LTCF, triggering implementation of surveillance interventions (see below). We further assumed a low baseline rate of subsequent SARS-CoV-2 introductions from the community, again calibrated to French data and depicting a situation of ongoing localized risk. See Supplementary information section I for more initialization details. Outbreaks were simulated over two weeks to evaluate short-term outbreak risk and immediate health-economic benefits of surveillance interventions.

**Surveillance interventions.** Surveillance interventions were implemented in response to the identified surge in nosocomial outbreak risk at simulation outset. We distinguish between *routine testing*, the targeted use of RT-PCR upon onset of COVID-19-like symptoms or admission of new patients into the LTCF; and *population screening*, the mass testing of entire populations (e.g. patients, staff) on selected dates. We assessed 27 surveillance interventions grouped into four categories: (i) routine testing, (ii) 1-round screening, (iii) routine testing + 1-round screening, and (iv) routine testing + 2-round screening (see list of interventions in Supplementary table S2). The latter two categories are defined as *multi-level surveillance interventions* that combine both screening and testing. Based on published estimates, diagnostic sensitivities of RT-PCR $s_{PCR}(t)$ and Ag-RDT $s_{RDT}(t)$ were assumed to vary with time since SARS-CoV-2 exposure $t$[42,43]. Ag-RDT was on average 73.5% as sensitive as RT-PCR, with greater sensitivity (87.5%) up to 7-days post-symptom onset and lower sensitivity (64.1%) thereafter (see sensitivity curves in Supplementary Fig. S5 and further methodological detail in Supplementary information section III). For diagnostic specificity, we assumed 99.7% for Ag-RDT and 99.9% for RT-PCR[43].

**Simulating counterfactual scenarios.** Surveillance interventions were applied retrospectively to daily outbreak data for precise estimation of intervention effects, using methods adapted from single-world counterfactual analysis (see Kaminsky et al.[44]). Counterfactual scenarios were simulated by (i) retrospectively isolating individuals who test positive for SARS-CoV-2 (assuming immediate isolation for Ag-RDT but a 24-h lag for RT-PCR, reflecting a delay between sample and result), and (ii) pruning transmission chains, (removing all transmission events originating from isolated individuals). Single-world matching facilitated estimation of marginal benefits of multi-level surveillance interventions, i.e. additional benefits of population screening relative to a baseline routine testing intervention already in place (illustrated in Fig. 5). Simulation of counterfactual scenarios is described further in Supplementary information section III. We simulated 100 counterfactual scenarios per intervention per outbreak, for $n = 43.7$ million simulations for estimation of surveillance efficacy, efficiency and cost-effectiveness.

**Surveillance outcomes: efficacy.** For each outbreak simulation, the cumulative number of nosocomial infections at two weeks, $I$, was calculated in the absence of surveillance interventions as $I_{baseline}$. For each counterfactual scenario, the number of infections averted by transmission chain pruning for each surveillance intervention was calculated as $I_{averted}$, so the adjusted incidence for each surveillance counterfactual was calculated as

$$I_{surveillance} = I_{baseline} - I_{averted} \qquad (1)$$

The *relative efficacy*, $E$, of each surveillance intervention for each counterfactual was calculated as the proportional reduction in $I$, given by

$$E = 1 - \frac{I_{surveillance}}{I_{baseline}} \qquad (2)$$

For multi-level interventions combining routine testing ("testing") and population screening ("screening"), *marginal relative efficacy of screening*, $E_m$, was calculated by excluding infections already averted due to testing,

$$E_m = 1 - \frac{I_{testing+screening}}{I_{testing}} \qquad (3)$$

**Surveillance outcomes: efficiency.** We calculated four measures of surveillance efficiency. First, *apparent efficiency*, $A$, was defined as perceived operational efficiency, calculated using the number of cases detected by surveillance ($D_{surveillance}$) as

$$A = \frac{D_{surveillance}}{n} \qquad (4)$$

where $n$ is the number of tests used.

Second, *real efficiency*, $R$, was defined as the relative health benefit resulting from intervention, calculated using the per-test number of infections averted as

$$R = \frac{I_{baseline} - I_{surveillance}}{n} \qquad (5)$$

Third, for multi-level interventions combining testing and screening, *marginal real efficiency of screening*, $R_m$, was calculated by excluding infections already averted and tests already used due to testing, given by

$$R_m = \frac{I_{testing} - I_{testing+screening}}{n_{screening}} \qquad (6)$$

Fourth, the *cost-effectiveness ratio*, CER, was defined as total surveillance costs per case averted, accounting for unit costs $c$ of routine testing ($c_{testing}$) and screening ($c_{screening}$),

$$CER = \frac{n_{testing} \times c_{testing} + n_{screening} \times c_{screening}}{I_{baseline} - I_{surveillance}} \qquad (7)$$

where we assumed the use of RT-PCR for routine testing at a baseline €50/test, and Ag-RDT for population screening at a baseline €5/test, similar to previous cost estimates for France and the UK[45,46]. Other outcomes evaluated to assess the performance of testing and screening interventions were true-positive rate (TPR), true-negative rate (TNR), negative predictive value (NPV) and positive predictive value (PPV).

**Statistics.** All surveillance outcomes are reported as means across $n = 10,000$ simulations (100 outbreaks × 100 surveillance runs) and were calculated in R software v3.6.0. 95% confidence intervals were calculated using bootstrap resampling with 100 replicates and normal approximation (R package *boot*).

**Reporting summary.** Further information on research design is available in the Nature Research Reporting Summary linked to this article.

## Data availability

Synthetic contact data used in CTCmodeler are available from DS within one month if requested for public research purposes. Outbreak datasets generated by CTCmodeler and surveillance outcome datasets resulting from the present study are available at https://github.com/drmsmith/agrdt/ [47].

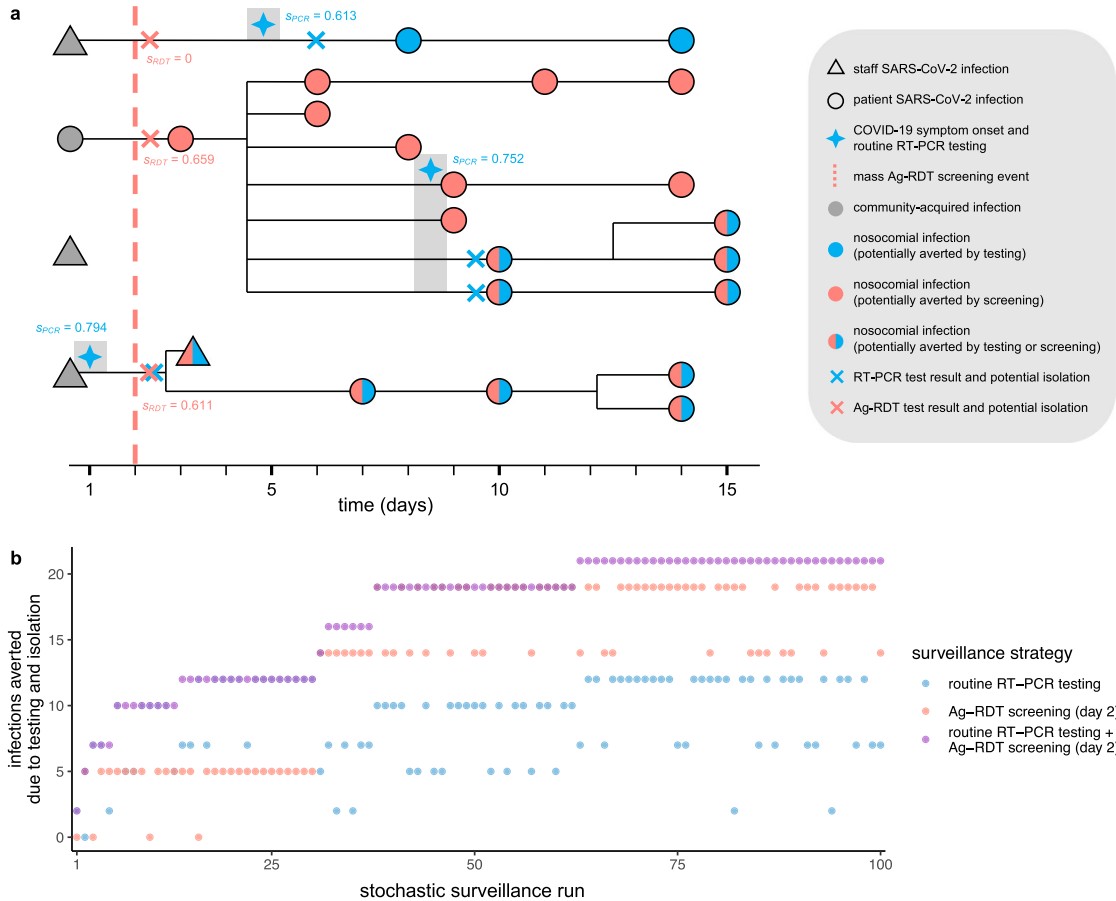

**Fig. 5 Surveillance interventions were applied retrospectively to simulated SARS-CoV-2 outbreaks, illustrated here using data from outbreak simulation #22 from LTCF 1. a** The SARS-CoV-2 transmission chain, with infections (shapes) transmitted from left to right following black lines. Of four community-onset infections (grey shapes) at simulation outset, three transmitted to other individuals in the LTCF, triggering a nosocomial outbreak. Routine RT-PCR testing was conducted upon COVID-19 symptom onset (blue four-pointed stars), with results and case isolation 24-h later (blue crosses). A population-wide Ag-RDT screening event was conducted on day 2 (red dashed line) with immediate results and isolation (red crosses). Test sensitivity— the probability of a positive test result and subsequent isolation—is given by $s$ adjacent to each test, as determined by infection age $t$ at the time of each test (see Supplementary Fig. S5). Nosocomial infections are coloured blue if potentially averted by routine testing, red if by screening, or both if by either. **b** Corresponding surveillance results from three selected surveillance interventions evaluated over 100 stochastic surveillance runs. The multi-level testing + screening intervention always averted at least as many infections as either individual intervention in the same run, demonstrating matching of "controlled" and "uncontrolled" epidemics across interventions, and its relevance for calculation of marginal benefits of multi-level interventions.

## Code availability

Code developed during the present study is available at https://github.com/drmsmith/agrdt/ [47].

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

## Acknowledgements

The work was supported directly by internal resources from the French National Institute for Health and Medical Research (Inserm), the Institut Pasteur, the Conservatoire National des Arts et Métiers, and the University of Versailles–Saint-Quentin-en-Yvelines/University of Paris-Saclay. This study received funding from the French Government's "Investissement d'Avenir" program, Laboratoire d'Excellence "Integrative Biology of Emerging Infectious Diseases" (Grant ANR-10-LABX-62- IBEID), the MODCOV project from the Fondation de France (Grant 106059) as part of the alliance framework "Tous unis contre le virus", the Université Paris-Saclay (AAP Covid-19 2020) and the French government through its National Research Agency project SPHINX-17-CE36-0008-01. DS is also supported by a Canadian Institutes of Health Research Doctoral Foreign Study Award (Funding Reference Number 164263).

## Author contributions

L.O. and L.T. contributed equally in conceiving the study, with contributions from J.R.Z., and supervising the work. A.D. simulated outbreaks. D.S. simulated surveillance, performed analyses and rendered figures. All authors contributed to interpretation of results. D.S. drafted the paper. All authors revised the paper and approved the final version.

## Competing interests

L.O. reports a research grant from Pfizer outside the submitted work. J.R.Z. has received funding from Pfizer and Merck Sharp and Dohme for a research project through his institution, outside of the submitted work. All other authors report no other conflicts of interest.

## Additional information

**the EMAE-MESuRS Working Group on Nosocomial SARS-CoV-2 Modelling**

Audrey Duval[1,2,4,8], Niels Hendrickx[1], Kévin Jean[3,6,7], Sofía Jijón[3,6], Ajmal Oodally[1,2,3], Lulla Opatowski[1,2], George Shirreff[1,2,3], David R. M. Smith [1,2,3,8 ✉], Cynthia Tamandjou[2] & Laura Temime[3,6]

[7]MRC Centre for Global Infectious Disease Analysis, Department of Infectious Disease Epidemiology, Imperial College London, London, UK.

