## [Peer Review File · Nature Communications]

Reviewers' Comments:

Reviewer #1:

Remarks to the Author:

An agent-based model was used to evaluate the effectiveness and cost-effectiveness of different testing strategies in long-term care facilities under three distinct levels of COVID-19 containment. Counterfactual analysis was conducted to evaluate the effectiveness of different control measures. This process involved pruning previously derived transmission chains. The use of the three levels of COVID containment illustrates how the vulnerability of a long-term care facility influences the choice of a potential screening strategy. The authors found that although combined testing and screening were extremely effective epidemiologically, they tended to be less cost-effective.

1. In the SI, histograms show the number of ensembles in which there were zero cases reported over two weeks. The percentage of ensembles with zero cases should be noted in the main text to better distinguish the overall reduction in the outbreak size.
2. I am concerned that the counterfactual analysis may overestimate the effectiveness of the interventions. The supplement states that the CTC modeler used to simulate the outbreaks provides a list of nosocomial transmission events of donor and recipient. However, it is unclear whether this list also includes exposures after infection. For example, suppose an agent is infected by the index agent but is later exposed to other infectious agents. For some control interventions (i.e. testing), removal of the index agent should not eliminate the whole chain of transmission. The reason being is that the infection in that one individual may essentially just be delayed, not eliminated.
3. The effect asymmetry in vaccine immunity among staff and patients would have on the effectiveness of these strategies in the different settings is unclear. Currently, it is assumed that vaccine immunity is homogeneous among patients and staff. Higher coverage among patients could suggest focusing testing more so on staff. Higher coverage among staff would reduce the extent of introduction into the facility. Understanding these factors in asymmetry in immunity levels would aid in guiding policy and could affect the cost-effectiveness presented in Figure 4, where different targets are screened.
4. The specificity of the tests is not considered in the analysis. The specificity of the tests is important since staff is being replaced upon a positive test. As false-positives during screening could lead to the introduction of disease after the screening process, decreasing the effectiveness.
5. Line 110 in the SI. The units of minutes should be indicated for the duration.
6. Line 111-112 in the SI. The probability of transmission is the product of the SARS-CoV-2 transmission rate per minute of infectious contact and the duration. Should this probability not be $1-(1-p)^d$? This difference would only have an effect for long durations of contact. Also, in the SI it is unclear what motivates the saturation of the probability of infection after 60 min.
7. Line 131 in the SI states "an isolation duration equivalent to symptom duration". I think this statement would be better phrased as "an isolation equivalent to the remaining duration of infection (i.e., duration of symptoms)". This adjustment is more explicit, as some individuals may still be infectious after most symptoms resolve.
8. The 20% level of immunity should be stated in the methods of the main text and just not in the SI. It is currently stated in one of the main Figures.
9. Line 332-333 in the SI. Missing the division term in the sum in order to compute the average relative sensitivity.
10. In the abstract, it is reported that a 4-5 day delay between screening is optimal. However, this assessment was done only by evaluating the time of a single second mass screening follow up test. If testing was conducted everyday, then that would be more effective in mitigating disease transmission. This result should be clarified in the abstract or alternatively examine different testing frequencies over the short period of time.

Reviewer #2:

Remarks to the Author:

This is an interesting simulation study that explores the effect of different intervention strategies (particularly focused on testing strategies) in long-term care facilities (LTCF). What sets this work apart from similar previous studies is that next to effectiveness in preventing transmissions,

testing strategies are also evaluated along the dimension of cost-effectiveness, i.e. number of infections prevented per thousands of Euros invested in testing. Overall, the manuscript is well written and the analysis results are easy to follow. However, the study has some major issues with respect to missing robustness tests and the surprising lack of considering vaccination coverage in more detail, which makes me wonder how relevant the findings might be in practice.

(1) Regarding the first major issue, I note that the study considers three different scenarios for control measures but that there is little added in terms of robustness tests beyond that. Note that also the effects of the individual control measures can be expected to come with confidence intervals, but I can accept that taking the upper estimates for intervention effectiveness within one scenario might lead one effectively to the scenario with more stringent control measures, but this should be discussed. The situation is less clear with two assumptions that appear to be key to me for the paper's claims, namely

(i) test sensitivities. False negative rates (for single, non-repeated testing) for PCR tests appear to be highly heterogeneous in the literature [1], varying between close to zero or almost 50%. Also the reported sensitivities of Ag-RDT vary widely and may strongly depend on patient characteristics (symptomatic/asymptomatic) and manufacturer.

(ii) test costs. PCR tests have the substantial advantage that they allow for pooling. I happen to know that for LTCFs in another European country with PCR-based screening strategies we are currently operating at unit costs of around 8-10€ per PCR test and this is expected to approach 5€/test, whereas the manuscript uses 50€/test. Conversely we estimate costs of Ag-RDT to be substantially above 10€/test (certainly higher than pooled PCR tests) whereas the manuscript assumes 5€/test. Naturally, costs might differ substantially across regions and heavily depend on previous investments in infrastructure and test logistics which is beyond the scope of this work.

Taken together, I would find it more appropriate if costs and sensitivities are not fixed but rather the authors could provide more insights into the conditions under which PCR or Ag-RDT tests are more cost-effective or useful than the other. For a given test of a, say, Ag-RDT test, can one define a break-even cost for PCR tests in terms of cost-effectiveness? Similar for sensitivities. Without such further analyses, it is questionable how the results presented here generalize to other LTCFs in different regions.

(2) As far as I see, the study includes vaccination coverage in the form of different degrees of immunization in the three control scenarios. Immunization rates of 20% and 50% are considered. This is surprising, as particularly European LTCFs can be expected to have a substantially higher vaccination coverage amongst residents while the coverage in the staff might be lower (or higher in case of mandatory vaccinations). One of the pressing questions of course is the point to which vaccination and testing need to exist side-by-side in LTCFs, which is not at all addressed in this manuscript. So given that most of the results reported in the paper have been measured at relatively low immunization, I wonder how relevant these findings are right now in practice and would have expected to see also results for higher vaccination coverages.

Some other minor issues:

The paper does not consider the burden associated with different testing technologies. PCR tests allow for sample collection by means of saliva (gargling tests) while swabs collected for Ag-RDT tests can lead to adherence problems in collected on a regular basis as they are not really comfortable to experience). One way around this is to use a nasopharyngeal swab which in turn reduces test sensitivity. This issue could be discussed in a bit more detail.

I was wondering whether the way that the results concerning cost-effectiveness are reported is misleading or not. Of course, higher incidences lead to a higher cost-effectiveness of testing. But given the outbreak sizes observed, maybe this is not something that one wants to optimize. Maybe cost-effectiveness is a more useful indicator when comparing the testing strategies only in scenarios with stronger control measures or higher vaccination coverage?

Reference:

[1] <https://journals.plos.org/plosone/article?id=10.1371/journal.pone.0242958>

REVIEWER COMMENTS

Reviewer #1 (Remarks to the Author):

An agent-based model was used to evaluate the effectiveness and cost-effectiveness of different testing strategies in long-term care facilities under three distinct levels of COVID-19 containment. Counterfactual analysis was conducted to evaluate the effectiveness of different control measures. This process involved pruning previously derived transmission chains. The use of the three levels of COVID containment illustrates how the vulnerability of a long-term care facility influences the choice of a potential screening strategy. The authors found that although combined testing and screening were extremely effective epidemiologically, they tended to be less cost-effective.

We thank the reviewer sincerely for their thoughtful comments and constructive feedback.

1. In the SI, histograms show the number of ensembles in which there were zero cases reported over two weeks. The percentage of ensembles with zero cases should be noted in the main text to better distinguish the overall reduction in the outbreak size.

The following text has been added to the caption for Figure 1 to better distinguish overall change in incidence across scenarios. We present the number of simulations with any transmission (instead of no transmission) to facilitate subsequent comparison with the number of simulations with increasing thresholds of cases (5, 10 and 25 cases).

The proportions of simulations with ≥ 1 cumulative nosocomial cases were 98%, 98% and 59% in LTCFs 1, 2 and 3, respectively; the proportions with ≥ 5 cumulative nosocomial cases were 91%, 73% and 2%; the proportions with ≥ 10 cumulative nosocomial cases were 82%, 55% and 0%; and the proportions with ≥ 25 cumulative nosocomial cases were 51%, 6% and 0%.

2. I am concerned that the counterfactual analysis may overestimate the effectiveness of the interventions. The supplement states that the CTC modeler used to simulate the outbreaks provides a list of nosocomial transmission events of donor and recipient. However, it is unclear whether this list also includes exposures after infection. For example, suppose an agent is infected by the index agent but is later exposed to other infectious agents. For some control interventions (i.e. testing), removal of the index agent should not eliminate the whole chain of transmission. The reason being is that the infection in that one individual may essentially just be delayed, not eliminated.

We thank the reviewer for raising this point, and agree with the argument that transmission pruning does not account for delayed infection (i.e. individuals whose infections were averted retrospectively, who could nonetheless become infected via a subsequent infectious contact). This could potentially have overestimated the efficacy of interventions, especially in lower control LTCFs where the virus circulates more.

However, we do not believe such a delayed infection effect has had a qualitative impact on our findings. Transmission pruning was applied identically across all interventions, and so any effect should in theory have an equal impact across all main outcomes and

stratifications thereof (e.g. testing vs. screening effectiveness, targeting patients vs. staff, testing with RT-PCR vs Ag-RDT).

We also do not expect that this effect would have had a substantive impact on higher control LTCFs, which had few infections in the absence of surveillance (mean 1.1 infections in LTCF 3 over a mean 5,740 person-days), translating to low risk of delayed infection from a subsequent contact in the event of isolation of an initially infectious contact.

We now highlight this methodological limitation in the Discussion (p. 13):

Third, counterfactual scenarios did not account for possible alternative infectors due to subsequent infectious exposures. For instance, an individual whose infection was averted due to isolation of their infector should nonetheless remain at risk of infection during subsequent contacts with other infectious individuals. This effect was likely negligible in higher-control LTCFs, where multiple acquisition routes are unlikely in the context of low nosocomial incidence, but may have resulted in overestimation of intervention efficacy in lower-control LTCFs. However, this should not have qualitatively changed our conclusions, as transmission chain pruning was conducted identically across interventions (e.g. testing vs. screening), and screening timing, test type (RT-PCR vs. Ag-RDT) and target (patients vs. staff).

3. The effect asymmetry in vaccine immunity among staff and patients would have on the effectiveness of these strategies in the different settings is unclear. Currently, it is assumed that vaccine immunity is homogeneous among patients and staff. Higher coverage among patients could suggest focusing testing more so on staff. Higher coverage among staff would reduce the extent of introduction into the facility. Understanding these factors in asymmetry in immunity levels would aid in guiding policy and could affect the cost-effectiveness presented in Figure 4, where different targets are screened.

We agree that asymmetric immunization could have important consequences on epidemiological dynamics and hence screening efficiency, with of course less immunized sub-populations being at higher risk for infection and hence priority targets for surveillance. We have conducted a new sensitivity analysis (Supplementary figure S4) evaluating how asymmetric vaccine coverage impacts SARS-CoV-2 outbreak risk (measured as mean R_0 at simulation outset for each LTCF, i.e. the average number of secondary infections resulting from index cases). This is introduced in Methods:

In a sensitivity analysis, we varied rates of immunizing seroprevalence from 0% to 100% across patients and staff to investigate potential epidemiological impacts of asymmetric immunization coverage.

We find that R_0 is overall substantially lower among staff index cases than patient index cases. We also find that increasing patient vaccination has a greater impact on reducing R_0 for patient index cases, while patient and staff vaccination have similar impacts for reducing overall R_0 for staff index cases. In higher-control LTCFs, changes in vaccination rates have comparatively limited impacts on reducing R_0 , which is already substantially reduced by alternative COVID-19 control measures. All of these findings are broadly consistent with previous results already discussed in the manuscript, such as

the relative contributions of patients and staff to SARS-CoV-2 transmission across LTCFs 1, 2 and 3. These findings are presented in Results:

In a sensitivity analysis evaluating outbreak risk across asymmetric levels of patient and staff vaccine coverage, patient immunization was more effective than staff immunization for preventing patients from transmitting (see Supplementary figure S4). Conversely, patient and staff immunization were similarly effective for preventing staff from transmitting. However, increasing immunization had a comparatively small impact on outbreak control in LTCFs with alternative control measures already in place (i.e. social distancing and face masks).

And the new figure is presented in the supplementary appendix:

Supplementary figure S4. The mean (range) of the number of secondary nosocomial infections caused by index cases (for simplicity, R_0), stratified by (A) patient index cases and (B) staff index cases. Control measures for each LTCF are the same as presented in Figure 1, except for vaccination: here, the proportion of patients immunized at simulation outset is varied along the x-axis, and the proportion of staff along the y-axis.

More generally, to put these findings in context, we now discuss in greater detail (p. 12) that the main goal of this study was detailed analysis of testing and screening interventions across selected risk scenarios. Detailed analysis of the efficacy and impacts of alternative (i.e. non-surveillance) control measures such as vaccination was beyond the scope of this work:

This work focused on detailed evaluation of surveillance interventions, explored in the context of three focal scenarios corresponding to varied adherence to standard COVID-19 prevention measures (social distancing, face masks, vaccination). Each scenario resulted in fundamentally different epidemic dynamics (exponential growth, linear growth and extinction), allowing us to

demonstrate how health-economic efficiency of surveillance varies with underlying nosocomial outbreak risk, while relative epidemiological efficacy is largely conserved. However, a detailed assessment of how different combinations of these other COVID-19 prevention measures impact outbreak risk, transmission dynamics, surveillance efficacy and surveillance efficiency was beyond the scope of this work. To take the example of vaccination, outbreak risk (and hence surveillance efficiency) depends on, among other variables: the particular vaccine(s) used; their efficacy for prevention of transmission and disease in the context of locally circulating variants; the distribution of the number of doses/boosters received across patients and staff; and associated rates of immune waning and breakthrough infection. In a supplementary analysis we show how outbreak risk in our simulated LTCFs varies across asymmetric levels of immunizing seroprevalence (Figure S4), which may in turn impact optimal targets for screening (e.g. patient screening should likely be prioritized in a facility with disproportionately low patient or high staff vaccine coverage). For real-world facilities, local outbreak risk must be continuously assessed using local and up-to-date demographic, epidemiological and immunological data.

4. The specificity of the tests is not considered in the analysis. The specificity of the tests is important since staff is being replaced upon a positive test. As false-positives during screening could lead to the introduction of disease after the screening process, decreasing the effectiveness.

We thank the reviewer for raising this point. The specificity of tests was indeed considered in the analysis, as evidenced by the simulated true negative rates in supplementary figure S10B, and corresponding specificity functions included in R code (see <https://github.com/drmsmith/agrdt>, functions.R, line 1200). Omission of an explicit statement about this from the text was an oversight on our part; we regret the error and have now included the following in the third Methods paragraph of the main text:

For diagnostic specificity, we assumed 99.7% for Ag-RDT and 99.9% for RT-PCR.(Brümmer et al. 2021)

5. Line 110 in the SI. The units of minutes should be indicated for the duration.

This has been added.

6. Line 111-112 in the SI. The probability of transmission is the product of the SARS-CoV-2 transmission rate per minute of infectious contact and the duration. Should this probability not be $1-(1-p)^d$? This difference would only have an effect for long durations of contact. Also, in the SI it is unclear what motivates the saturation of the probability of infection after 60 min.

We thank the reviewer for this helpful suggestion, which we agree is a more elegant means to model saturating transmission probability at high contact duration than our step-wise cut-off. Although a cut-off at 1 hour may lack obvious biological realism, we lacked data to inform a more evidence-based alternative assumption, and believe that changing the shape of this curve would have a negligible impact on epidemic outcomes and surveillance results. We also agree with the reviewer that changing the shape should only have an impact on particularly long durations of contact. In our simulations, the

median duration of an infectious contact was 28.5 minutes, and approximately 80% of infectious contacts were less than 1 hour in length, so we do not believe that adjusting this would have an appreciable impact on transmission. This is now mentioned in the Discussion:

Further, the assumption that transmission risk saturates after 1 hour of infectious contact may lack biological realism, but is unlikely to have substantially affected transmission dynamics (approximately 80% of contacts were <60m in duration; median duration = 28.5m).

7. Line 131 in the SI states "an isolation duration equivalent to symptom duration". I think this statement would be better phrased as "an isolation equivalent to the remaining duration of infection (i.e., duration of symptoms)". This adjustment is more explicit, as some individuals may still be infectious after most symptoms resolve.

Thank you for this suggestion, the sentence has been updated.

8. The 20% level of immunity should be stated in the methods of the main text and just not in the SI. It is currently stated in one of the main Figures.

The following highlighted text has been added to the first Methods paragraph:

These include: (i) a patient social distancing intervention (cancellation of social activities; see Supplementary figure S1), (ii) mandatory face masks among patients and staff (80% reduction in transmission rates), and (iii) partial vaccination of patients and staff (50% immunizing seroprevalence at simulation outset, compared to an assumed 20% baseline in scenarios without vaccination).

9. Line 332-333 in the SI. Missing the division term in the sum in order to compute the average relative sensitivity.

Thank you for identifying this, the summation terms have been updated.

10. In the abstract, it is reported that a 4-5 day delay between screening is optimal. However, this assessment was done only by evaluating the time of a single second mass screening follow up test. If testing was conducted everyday, then that would be more effective in mitigating disease transmission. This result should be clarified in the abstract or alternatively examine different testing frequencies over the short period of time.

This has been clarified using the highlighted text:

For the latter, a delay of 4-5 days between the two screening rounds was optimal for transmission prevention.

Reviewer #2 (Remarks to the Author):

This is an interesting simulation study that explores the effect of different intervention strategies (particularly focused on testing strategies) in long-term care facilities (LTCF). What sets this work apart from similar previous studies is that next to effectiveness in preventing transmissions, testing strategies are also evaluated along the dimension of cost-effectiveness, i.e. number of infections prevented per thousands of Euros invested in testing. Overall, the manuscript is well written and the analysis results are easy to follow. However, the study has some major issues with respect to missing robustness tests and the surprising lack of considering vaccination coverage in more detail, which makes me wonder how relevant the findings might be in practice.

We thank the reviewer for their thoughtful analysis and helpful comments about our work.

(1) Regarding the first major issue, I note that the study considers three different scenarios for control measures but that there is little added in terms of robustness tests beyond that. Note that also the effects of the individual control measures can be expected to come with confidence intervals, but I can accept that taking the upper estimates for intervention effectiveness within one scenario might lead one effectively to the scenario with more stringent control measures, but this should be discussed.

We thank the reviewer for raising this and certainly agree that SARS-CoV-2 control measures come with a wide degree of uncertainty. However, we highlight that the goal of this work was only to evaluate surveillance interventions (not control measures). Our motivation for including differing levels of COVID-19 control was to demonstrate how surveillance outcomes may or may not vary across risk scenarios: a ‘high-risk’ scenario with exponential epidemic growth, a ‘moderate risk’ scenario with linear growth, and a ‘low risk’ scenario with epidemic extinction. Any conclusions about other interventions – e.g. their epidemiological impact and associated uncertainty – are outside the scope of the present work. To clarify this point, we have now included a new discussion paragraph highlighting this context:

This work focused on detailed evaluation of surveillance interventions, explored in the context of three focal scenarios corresponding to varied adherence to standard COVID-19 prevention measures (social distancing, face masks, vaccination). Each scenario resulted in fundamentally different epidemic dynamics (exponential growth, linear growth and extinction), allowing us to demonstrate how health-economic efficiency of surveillance varies with underlying nosocomial outbreak risk, while relative epidemiological efficacy is largely conserved. However, a detailed assessment of how different combinations of these other COVID-19 prevention measures impact outbreak risk, transmission dynamics, surveillance efficacy and surveillance efficiency was beyond the scope of this work.

The situation is less clear with two assumptions that appear to be key to me for the paper's claims, namely

(i) test sensitivities. False negative rates (for single, non-repeated testing) for PCR tests appear to be highly heterogeneous in the literature [1], varying between close to zero or almost 50%. Also the reported sensitivities of Ag-RDT vary widely and may strongly depend on patient characteristics (symptomatic/asymptomatic) and manufacturer.

By using estimates of test sensitivity from systematic reviews and meta-analyses (Kucirka et al. *Ann Intern Med* 2020 for RT-PCR, and for Ag-RDT both Dinnes et al. *Cochrane Database Syst Rev* 2021, and Brümmer et al. *medRxiv* 2021), we sought to use the most widely representative data available. Crucially, these meta-analyses are among the very few that have estimated test sensitivity as a function of time since infection, and suggest that a great share of heterogeneity in false-negative rates (FNRs) can probably be explained by variability in testing timing (e.g. false-negatives due to testing early in infection, concomitant with low viral load). In our view, this time-varying nature is the most important element to consider regarding test sensitivity when evaluating the efficacy of SARS-CoV-2 surveillance interventions. Indeed, a key limitation of some contemporary models assessing SARS-CoV-2 surveillance interventions is the assumption of time-invariant test sensitivity (e.g. See et al. *Clin Infect Dis* 2021). We have added a sentence in the discussion highlighting this strength.

This work was further strengthened through use of time-varying, test-specific diagnostic sensitivity (as opposed to time-invariant estimates often assumed in other work), facilitating assessment of optimal timing for multi-round screening.

Nonetheless, we absolutely agree that there is still unexplained heterogeneity across published studies, that some tests are more sensitive than others, and that considering alternative estimates of test sensitivity would have important consequences for our surveillance outcomes. This was our motivation for including several sensitivity analyses related to test sensitivity (Figure S8), including:

- Ag-RDT B: an alternative time-varying sensitivity curve for Ag-RDT
- Perfect sensitivity: 0% FNRs for both RT-PCR and Ag-RDT
- Uniform sensitivity: 30% FNR for RT-PCR and 46% for Ag-RDT

We note that these FNR values broadly cover the range proposed by the reviewer. These analyses demonstrated that screening using perfect tests would prevent >95% of nosocomial transmission secondary to a surge in outbreak risk, while repeated screening using tests with uniform sensitivity would be most effective were the sequential rounds of testing to be conducted without delay. This latter finding further reinforces the importance of accounting for time-varying sensitivity curves: only when this is taken into account do we find that an intermediate delay of 4-5 days in second-round testing is optimal, accounting for the fact that false-negative results are highly probable among nascent infections screened during the first round.

Finally, we believe our use of best available estimates for time-varying diagnostic sensitivity accounts for supposed higher sensitivity in symptomatic vs. non-symptomatic infections, which may be better explained by testing timing than by true differences between symptomatic and asymptomatic infections (i.e. lower sensitivity among “asymptomatic” infections when they also include pre-symptomatic infections). To our knowledge the literature suggests that, over the full course of infection, there is no significant difference in average test sensitivity across symptomatic (including pre-symptomatic) and truly asymptomatic infections. (Ladhani et al. *EClinicalMedicine* 2020 ; Lee et al. *JAMA Intern Med* 2020; Singanayagam et al. *Euro Surveill* 2020; Brümmer et al. *medRxiv* 2021)

This is now made explicit in the supplementary appendix for RT-PCR:

Consistent with findings from the literature, we assumed no difference in diagnostic sensitivity for symptomatic (including pre-symptomatic) and asymptomatic (including pre-asymptomatic) infections.[22–24]

and for Ag-RDT in the context of the meta-analysis from Brümmer et al.:

They found no statistical difference in Ag-RDT sensitivity between symptomatic and asymptomatic patients.

(ii) test costs. PCR tests have the substantial advantage that they allow for pooling. I happen to know that for LTCFs in another European country with PCR-based screening strategies we are currently operating at unit costs of around 8-10€ per PCR test and this is expected to approach 5€/test, whereas the manuscript uses 50€/test. Conversely we estimate costs of Ag-RDT to be substantially above 10€/test (certainly higher than pooled PCR tests) whereas the manuscript assumes 5€/test. Naturally, costs might differ substantially across regions and heavily depend on previous investments in infrastructure and test logistics which is beyond the scope of this work.

Taken together, I would find it more appropriate if costs and sensitivities are not fixed but rather the authors could provide more insights into the conditions under which PCR or Ag-RDT tests are more cost-effective or useful than the other. For a given test of a, say, Ag-RDT test, can one define a break-even cost for PCR tests in terms of cost-effectiveness? Similar for sensitivities. Without such further analyses, it is questionable how the results presented here generalize to other LTCFs in different regions.

We are appreciative of this great suggestion and absolutely agree: we have now extended our cost-effectiveness analysis to cover a large range of potential costs of both Ag-RDT and RT-PCR testing (from €1 to €100 per test for each), instead of considering only a select few potential costs. This is evidenced in a new Figure 5 (the previous Figure 5 has been moved to the supplementary appendix):

Figure 5. Cost-effectiveness ratios for a highly epidemiologically effective surveillance strategy (routine RT-PCR testing + 2-round Ag-RDT screening on days 1 and 5), estimated as testing unit costs per infection averted while varying unit costs for RT-PCR tests (x-axis) and Ag-RDT tests (y-axis). Baseline assumptions underlying simulations include: “low” community SARS-CoV-2 incidence; time-varying Ag-RDT sensitivity relative to RT-PCR (Ag-RDT A); and screening interventions that target all patients and staff in the LTCF.

Cost-effectiveness results are now discussed in this new context as follows:

Cost-effectiveness ratios of surveillance interventions varied by orders of magnitude across LTCFs (Figure 5). In LTCF 1, assuming baseline costs of €50/RT-PCR test and €5/Ag-RDT test, routine RT-PCR testing + 2-round Ag-RDT screening cost €469 (€462–€478)/case averted, with similar estimates for 1-round screening (Figure S12). In LTCF 2, the same intervention cost €1,180 (€1,166–€1,200)/case averted, and in LTCF 3 €11,112 (€10,825–€11,419)/case averted. Overall, for this combined strategy of routine testing and reactive screening, cost-effectiveness ratios were more sensitive to costs of Ag-RDT screening tests than routine RT-PCR tests (Figure 5). At a fixed €50/RT-PCR test in the high-control LTCF 3, cost-effectiveness ratios were approximately €16,000 per case averted at €10/Ag-RDT test, €30,000 per case averted at €25/Ag-RDT test, and €54,000 per case averted at €50/Ag-RDT test. Conversely, in the low-control LTCF, cost-effectiveness ratios remained below €5,000 per case averted up to €100/Ag-RDT test. When reactive Ag-RDT screening and routine RT-PCR testing were considered separately as independent strategies, routine testing was always more cost-effective than reactive screening per € spent on surveillance costs (supplementary Figure S13).

(2) As far as I see, the study includes vaccination coverage in the form of different degrees of immunization in the three control scenarios. Immunization rates of 20% and 50% are considered. This is surprising, as particularly European LTCFs can be expected to have a substantially higher vaccination coverage amongst residents while the coverage in the staff might be lower (or higher in case of mandatory vaccinations). One of the pressing questions of course is the point to which vaccination and testing need to exist side-by-side in LTCFs, which is not at all addressed in this manuscript. So given that most of the results reported in the paper have been measured at relatively low immunization, I wonder how relevant these findings are right now in practice and would have expected to see also results for higher vaccination coverages.

We agree that varying levels of immunization could have important impacts on epidemiological dynamics and hence screening efficiency, with of course less immunized facilities or sub-populations being at higher risk for infection and hence priority targets for surveillance.

First, we emphasize that our simulated vaccine intervention assumes **50% immunizing seroprevalence and not 50% “vaccine coverage”**, the key difference being that coverage does not account for imperfect efficacy, immune waning and vaccine-escape variants whereas immunizing seroprevalence does. Given available VE estimates for the current dominant variant (Delta) and vaccine (BNT162b2) in Europe, we believe that our “vaccine” scenario is an appropriate representation of an average facility in Europe with high vaccine coverage. This distinction is now clarified in the methods:

A range of *COVID-19 containment measures* were built into the model. These include: (i) a patient social distancing intervention (cancellation of social activities; see Supplementary figure S1), (ii) mandatory face masks among patients and staff (80% reduction in transmission rates), and (iii) imperfect vaccination of patients and staff (50% immunizing seroprevalence at simulation outset, compared to an assumed 20% baseline in scenarios without vaccination). This value is consistent with an estimated 53% efficacy of the mRNA BNT162b2 vaccine against infection with the Delta variant four months from second dose.[17]

Second, following the reviewer’s feedback, we have conducted a new sensitivity analysis (Supplementary figure S4) evaluating how asymmetric vaccine coverage impacts SARS-CoV-2 outbreak risk (measured as mean R_0 at simulation outset for each LTCF, i.e. the average number of secondary infections resulting from index cases). This is introduced in Methods:

In a sensitivity analysis, we varied rates of immunizing seroprevalence from 0% to 100% across patients and staff to investigate potential epidemiological impacts of asymmetric immunization coverage.

We find that R_0 is overall substantially lower among staff index cases than patient index cases. We also find that increasing patient vaccination has a greater impact on reducing R_0 for patient index cases, while patient and staff vaccination have similar impacts for reducing overall R_0 for staff index cases. In higher-control LTCFs, changes in vaccination rates have comparatively limited impacts on reducing R_0 , which is already substantially reduced by alternative COVID-19 control measures. All of these findings are broadly consistent with previous results already discussed in the manuscript, such as the relative contributions of patients and staff to SARS-CoV-2 transmission across LTCFs 1, 2 and 3. These findings are presented in Results:

In a sensitivity analysis evaluating outbreak risk across asymmetric levels of patient and staff vaccine coverage, patient immunization was more effective than staff immunization for preventing patients from transmitting (see Supplementary figure S4). Conversely, patient and staff immunization were similarly effective for preventing staff from transmitting. However, increasing immunization had a comparatively small impact on outbreak control in LTCFs with alternative control measures already in place (i.e. social distancing and face masks).

And the new figure is presented in the supplementary appendix:

Supplementary figure S4. The mean (range) of the number of secondary nosocomial infections caused by index cases (for simplicity, R_0), stratified by (A) patient index cases and (B) staff index cases. Control measures for each LTCF are the same as presented in Figure 1, except for vaccination: here, the proportion of patients immunized at simulation outset is varied along the x-axis, and the proportion of staff along the y-axis.

More generally, to put these findings in context, we now discuss in greater detail (p. 12) that the main goal of this study was detailed analysis of testing and screening interventions across selected risk scenarios. Detailed analysis of the efficacy and impacts of alternative (i.e. non-surveillance) control measures such as vaccination was beyond the scope of this work:

This work focused on detailed evaluation of surveillance interventions, explored in the context of three focal scenarios corresponding to varied adherence to standard COVID-19 prevention measures (social distancing, face masks, vaccination). Each scenario resulted in fundamentally different epidemic dynamics (exponential growth, linear growth and extinction), allowing us to demonstrate how health-economic efficiency of surveillance varies with underlying nosocomial outbreak risk, while relative epidemiological efficacy is largely conserved. However, a detailed assessment of how different combinations of these other COVID-19 prevention measures impact outbreak risk, transmission dynamics, surveillance efficacy and surveillance efficiency was beyond the scope of this work. To take the example of vaccination, outbreak risk (and hence surveillance efficiency) depends on, among other variables: the particular vaccine(s) used; their efficacy for prevention of transmission and disease in the context of locally circulating variants; the distribution of the number of doses/boosters received across patients and staff; and associated rates of immune waning and breakthrough infection. In a supplementary analysis we show how outbreak risk in our simulated LTCFs varies across asymmetric levels of immunizing seroprevalence (Figure S4), which may in turn impact optimal targets for screening (e.g. patient screening should likely be prioritized in a facility with disproportionately low patient or high staff vaccine coverage). For real-world facilities, local outbreak risk must be continuously assessed using local and up-to-date demographic, epidemiological and immunological data.

Some other minor issues:

The paper does not consider the burden associated with different testing technologies. PCR tests allow for sample collection by means of saliva (gargling tests) while swabs collected for Ag-RDT tests can lead to adherence problems in collected on a regular basis as they are not really comfortable to experience). One way around this is to use a nasopharyngeal swab which in turn reduces test sensitivity. This issue could be discussed in a bit more detail.

We agree that different testing technologies have different costs and impose heterogeneous occupational burden. We have added the following sentence to the discussion, including a comment on group testing as previously mentioned by the reviewer:

Decision-makers may also have a wide variety of specific tests and manufacturers to choose from, including tests with heterogeneous sampling techniques (e.g. nasopharyngeal swabs vs. saliva or pharynx gargle samples), with potential consequences for surveillance costs, efficacy, compliance, and occupational burden. (Note that the RT-PCR and Ag-RDT sensitivity curves used in the present work represent average results across a range of different tests used on upper respiratory specimens.) In particular, group testing (sample pooling) may be an efficient means of surveillance in low prevalence settings, reducing overall testing costs.[16]

We also agree that screening interventions can have compliance issues, especially in the case of routine screening (e.g. once or several times weekly for an indefinite time period, as evaluated in previous modelling studies). However, we believe that adherence

problems are less relevant in our study, as we explicitly evaluate only one or two rounds of reactive screening in the context of a surge in outbreak risk.

I was wondering whether the way that the results concerning cost-effectiveness are reported is misleading or not. Of course, higher incidences lead to a higher cost-effectiveness of testing. But given the outbreak sizes observed, maybe this is not something that one wants to optimize. Maybe cost-effectiveness is a more useful indicator when comparing the testing strategies only in scenarios with stronger control measures or higher vaccination coverage?

We agree that the original Figure 5 was potentially misleading and did not best convey the message we were trying to communicate. We have now restructured the cost-effectiveness results to focus on the most epidemiologically effective scenario (routine testing + 2-round Ag-RDT screening on days 1 and 5) while varying Ag-RDT and RT-PCR unit costs from €1 to €100 (see new Figure 5). The main conclusions of this analysis are that cost-effectiveness scales heavily with underlying outbreak risk, and that cost-effectiveness is more sensitive to Ag-RDT costs than RT-PCR costs (already pasted above in response to the previous comment about testing costs). In this new Figure, it is now clearly visible that surveillance is highly cost-effective in low-control scenarios even at very high testing costs, suggesting that cost-effectiveness considerations are comparatively much more important in high-control settings, as the reviewer suggests.

We have moved the original Figure 5 to the supplement, emphasizing that cost-effectiveness estimates are overall similar between 1- and 2-round screening interventions regardless of testing unit costs. Finally, we have included one more additional cost-effectiveness plot evaluating routine testing and 1-round screening in isolation as independent interventions. We found that, in head-to-head comparison, routine testing was a more cost-effective investment than reactive screening:

Supplementary figure S13. Cost-effectiveness ratios for routine RT-PCR (left, strategy #1 from Supplementary table S2) and 1-round Ag-RDT screening on day 1 (right, strategy #2 from Supplementary table S2) as a function of testing unit costs.

UPDATED REFERENCES REFLECTING NEWLY PUBLISHED LITERATURE

We have included the following updates to the manuscript to reflect new studies published since original submission of our manuscript.

1. Breakthrough infections due to Delta variant (Pouwels et al. Nature Med 2021)

LTCFs globally report instances of breakthrough infection and ensuing transmission among immunized staff and residents, notably due to variants of concern like B.1.1.7 (Alpha), B.1.351 (Beta) and B.1.617.2 (Delta), which may partly escape vaccine-induced immunity relative to wild type.[5–8]

2. Estimated 53% efficacy of BNT162b2 vaccination against infection with Delta variant 4 months after second dose (Tartof et al. Lancet 2021)

This value is consistent with an estimated 53% efficacy of the mRNA BNT162b2 vaccine against infection with the Delta variant four months from second dose.[18]

3. Predicted high efficacy of PPE for preventing healthcare personnel from acquiring SARS-CoV-2 during vaccination campaigns, interpreted as potential evidence that PPE may equally protect healthcare workers from infection during screening campaigns (Procter et al. BMC Med 2021)

For instance, healthcare workers that conduct screening inevitably come into contact with many individuals, potentially creating new opportunities for transmission. This risk may be mitigated through appropriate use of PPE during screening,[40] and is not relevant if our results are interpreted in the context of self-administered auto-tests.

Reviewers' Comments:

Reviewer #1:

Remarks to the Author:

The authors have sufficiently addressed my prior comments

Reviewer #2:

Remarks to the Author:

I thank the authors for addressing all comments in a diligent and thorough way and have no further comments.

Reviewer #1 (Remarks to the Author):

The authors have sufficiently addressed my prior comments

Reviewer #2 (Remarks to the Author):

I thank the authors for addressing all comments in a diligent and thorough way and have no further comments.

We thank the reviewers for their time.